# Genomic and Glycolytic Entropy Are Reliable Radiogenomic Heterogeneity Biomarkers for Non-Small Cell Lung Cancer

**DOI:** 10.3390/ijms24043988

**Published:** 2023-02-16

**Authors:** Yu-Hung Chen, Kun-Han Lue, Chih-Bin Lin, Kuang-Chi Chen, Sheng-Chieh Chan, Sung-Chao Chu, Bee-Song Chang, Yen-Chang Chen

**Affiliations:** 1Department of Nuclear Medicine, Hualien Tzu Chi Hospital, Buddhist Tzu Chi Medical Foundation, Hualien 97002, Taiwan; 2School of Medicine, College of Medicine, Tzu Chi University, Hualien 97004, Taiwan; 3Department of Medical Imaging and Radiological Sciences, Tzu Chi University of Science and Technology, Hualien 97005, Taiwan; 4Department of Internal Medicine, Hualien Tzu Chi Hospital, Buddhist Tzu Chi Medical Foundation, Hualien 97002, Taiwan; 5Department of Medical Informatics, Tzu Chi University, Hualien 97004, Taiwan; 6Department of Hematology and Oncology, Hualien Tzu Chi Hospital, Buddhist Tzu Chi Medical Foundation, Hualien 97002, Taiwan; 7Department of Cardiothoracic Surgery, Hualien Tzu Chi Hospital, Buddhist Tzu Chi Medical Foundation, Hualien 97002, Taiwan; 8Department of Anatomical Pathology, Hualien Tzu Chi Hospital, Buddhist Tzu Chi Medical Foundation, Hualien 97002, Taiwan

**Keywords:** tumor heterogeneity, whole exome sequencing, ^18^F-FDG PET, molecular imaging, lung cancer

## Abstract

Radiogenomic heterogeneity features in ^18^F-fluorodeoxyglucose positron emission tomography (^18^F-FDG PET) have become popular in non-small cell lung cancer (NSCLC) research. However, the reliabilities of genomic heterogeneity features and of PET-based glycolytic features in different image matrix sizes have yet to be thoroughly tested. We conducted a prospective study with 46 NSCLC patients to assess the intra-class correlation coefficient (ICC) of different genomic heterogeneity features. We also tested the ICC of PET-based heterogeneity features from different image matrix sizes. The association of radiogenomic features with clinical data was also examined. The entropy-based genomic heterogeneity feature (ICC = 0.736) is more reliable than the median-based feature (ICC = −0.416). The PET-based glycolytic entropy was insensitive to image matrix size change (ICC = 0.958) and remained reliable in tumors with a metabolic volume of <10 mL (ICC = 0.894). The glycolytic entropy is also significantly associated with advanced cancer stages (*p* = 0.011). We conclude that the entropy-based radiogenomic features are reliable and may serve as ideal biomarkers for research and further clinical use for NSCLC.

## 1. Introduction

Lung cancer is among the most prevalent cancers worldwide, with 85% of lung cancer cases presenting with non-small cell lung cancer (NSCLC) histopathology [1]. Apart from histopathology, novel gene tests enable a more detailed cancer sub-classification according to driver gene mutations or the mutation burden [2]. The sub-classification based on driver gene mutations and mutation burden can guide targeted therapies such as those using tyrosine kinase and immune checkpoint inhibitors. Targeted therapeutics have improved survival outcomes of NSCLC compared to traditional chemotherapies [2,3]. Therefore, NSCLC is a heterogeneous disease, and gene testing has become pivotal in its management [2]. ^18^F-fluorodeoxyglucose positron emission tomography (^18^F-FDG PET) is an important imaging tool for assessing the extent of NSCLC. The metabolic radiomics derived from ^18^F-FDG PET feature tumor glycolytic activity and heterogeneity. Higher ^18^F-FDG PET-derived glycolytic activity and heterogeneity are associated with regional nodal metastasis and unfavorable survival outcomes. These features also reportedly predict pathological response after neoadjuvant chemoradiotherapy in patients with NSCLC [4,5,6]. The combination of gene tests and metabolic radiomics of ^18^F-FDG PET (radiogenomics) has gained increasing attention in NSCLC research [7,8].

In lung cancer, tumors evolve and accumulate sufficient mutations to allow dissemination; thus, the metastasis competent subclone arises at a late stage of tumor progression [9]. Hence, tumors accumulating more heterogeneous subclones are more likely to disseminate. Therefore, higher tumor heterogeneity is associated with a less favorable prognosis [9,10,11]. Genomics enables the quantification of genomic heterogeneity. ^18^F-FDG PET-derived radiomic features are able to reflect the heterogeneity of tumor glycolysis. Different genomic heterogeneity features, such as mutant-allele tumor heterogeneity (MATH, a median-based heterogeneity feature) and Shannon entropy (an entropy-based heterogeneity feature), have been reported to correlate with prognosis or glycolytic heterogeneity [12,13]. However, there is currently no consensus on the optimal population allele frequency (PAF) cutoff for somatic mutation calls in tumor-only tests [14,15]. Whether genomic heterogeneity features remain reliable when different PAFs are applied has not been thoroughly studied. On the other hand, the reliability of ^18^F-FDG PET-derived radiomics for lung cancer has been vigorously investigated. Features such as first-order entropy have been proven to be insensitive to respiratory motion and reproducible in different reconstruction or segmentation methods [16,17,18]. With the advent of novel digitalized PET hardware and advanced reconstruction algorithms, the image contrast and resolution have substantially improved, allowing larger image matrix sizes [19]. However, few studies have investigated whether changes in matrix size affects the quantification of PET-derived radiomics [20]. Radiomic features should be stable in different matrix sizes so that the biomarkers can retain their clinical values in the era of digitalized PET units.

Therefore, this prospective study aimed to investigate and compare the reliability of two commonly used genomic heterogeneity features (MATH and Shannon entropy) using different PAF cutoffs. We also analyzed the reliability of digitalized ^18^F-FDG PET-derived radiomics in different image matrix sizes. Finally, we correlated the genomic and radiomic biomarkers with clinical data from patients with NSCLC.

## 2. Results

### 2.1. Patient Characteristics

We enrolled 46 patients for the analysis. Twenty patients (43.5%) were diagnosed with stage IV disease. Whole exome sequencing (WES) specimens were obtained from surgical specimens in 27 (58.7%) patients and biopsies from 19 (41.3%) patients. The median tumor content of the WES specimens was 50%, with an interquartile range (IQR) of 32.5%. The means ± standard deviation (SD) of the ^18^F-FDG PET-based maximum standardized uptake value (SUV_max_), metabolic tumor volume (MTV), total lesion glycolysis (TLG) and entropy derived from standard image matrix size were 11.9 ± 7.01, 29.5 ± 55.15, 291.1 ± 646.69 and 4.2 ± 0.84, respectively. Regarding the genomic features, the means ± SD of tumor mutation burden from PAF cutoffs of 1% (TMBc), MATHc and Shannon entropy.c were 18.6 ± 8.91, 47.1 ± 14.54 and 2.8 ± 0.17, respectively. When using a PAF of 0.01% as the cutoff, the TMBm, MATHm and Shannon entropy.m were 12.0 ± 8.87, 55.5 ± 12.97 and 2.6 ± 0.24, respectively. We detected EGFR mutations in 16 patients (34.8%). BRAF and KRAS mutations were detected in one (2.2%) and two (4.3%) patients, respectively. Additionally, two patients (4.3%) had ALK overexpression. Otherwise, no ROS-1 rearrangement or MET mutations were detected.

### 2.2. Reliability of Radiogenomic Features

We assessed the reliability of radiogenomic features using the intraclass correlation coefficient (ICC), and Figure 1 presents the results. TMB showed excellent consistency under different PAF cutoff values. Regarding genomic heterogeneity, Shannon entropy was moderately reliable (ICC = 0.736), whereas MATH showed poor reliability (ICC = −0.416). Our study’s intensity, volumetric and heterogeneity radiomic features showed excellent reliability under different image matrix sizes (ICC > 0.9). We further performed subgroup analyses to examine how tumor size affects the reliability of radiomic features. Although the radiomic features generally showed inferior ICCs in patients with an MTV < 10 mL, the ICCs still showed good to excellent reliability.

### 2.3. Correlation of Radiogenomics and Clinical Characteristics

We assessed the Pearson’s correlation between genomic and radiomic features (Figure 2). Glycolytic imaging entropy was significantly correlated with Shannon entropy.c (r = 0.31, *p* = 0.036) and Shannon entropy.m (r = 0.32, *p* = 0.030). The PD-L1 expression in tumor cells (TC) did not significantly correlate with any radiogenomic features.

We also compared the radiogenomic features between patients with stages I–III and IV disease (Figure 3). Genomic heterogeneity did not differ significantly between patients with locoregional and disseminated diseases. Nevertheless, patients with stage IV disease showed significantly higher primary tumor glycolytic heterogeneity, intensity and volume. Furthermore, a significantly higher TMB was observed in ever-smokers (Appendix A). Otherwise, the distribution of radiogenomic features did not differ among the clinical characteristics, including smoking status, histology and actionable EGFR status.

The median follow-up of our patients was 13.1 m (1.8–26.7 m). Eleven (23.9%) patients died during follow-up. The association of overall survival (OS) with radiogenomic features is summarized in Table 1. Our preliminary survival analysis showed that a higher TMB, SUV_max_ and entropy derived from reduced matrix-size images were significantly associated with unfavorable OS. Additionally, the entropy derived from standard matrix-size images showed a trend toward a poor OS.

## 3. Discussion

With increasing knowledge on the molecular and genetic backgrounds of NSCLC, radiogenomics has become more important in the research and clinical applications of this disease [7]. Many quantitative radiogenomic features have emerged as cancer outcome biomarkers [7,12,13]. However, ensuring feature reliability is a major consideration in generalizing radiogenomic research and clinical implementation [7,18,21,22]. Our study found that MATH is not a reliable genomic heterogeneity feature, whereas Shannon entropy showed moderate reliability. Additionally, the glycolytic heterogeneity feature, entropy, along with the intensity (SUV_max_) and volumetric (MTV and TLG) PET-derived features, showed excellent reliability in terms of image matrix size change.

In our study, TMB and genomic heterogeneity features were derived from tumor-only gene sequencing data. The traditional methods of somatic mutation calling were performed on tumors and matching normal DNA [23]. However, this method requires sequencing matching normal tissue, and thus doubles the cost. The modern tumor-only methodology is also feasible and has been adopted in commercialized clinical sequencing services [15]. Instead of subtracting germline mutations by matching the sequencing data from normal tissue, the tumor-only method requires the PAF database to subtract germline mutation. Different PAF cutoffs have been proposed for synthesizing genomic features [14,15]. Gene sequencing-derived features are mainly used as biomarkers for clinical outcomes; therefore, the features should remain reliable when applying different PAF cutoffs, or the functional relationship between the features and clinical information may be lost. We found MATH to be an unreliable genomic heterogeneity feature according to our study results (ICC = −0.416). The MATH calculation is based on the median and median absolute deviation (MAD) of the mutant allele fraction (MAF). In patients with bimodally distributed MAF, the median value may show a more apparent shift when applying different PAF cutoffs than in patients with a single-peaked MAF distribution pattern (Appendix A). Therefore, MATH may vary substantially between the two PAF cutoffs. In contrast, entropy-based genomic features measure the general uncertainty distribution of MAF. Therefore, as long as the general distribution pattern of MAF remains unchanged, Shannon entropy would not shift drastically (Appendix A).

We also analyzed driver gene mutations in our study population. Actionable EGFR mutations were detected in 34.8% of our cohort, which is slightly lower than the reported prevalence of EGFR mutation in the Asian population [3], which may be explained by the male predominance (67.4%) in our cohort. Besides the EGFR mutation, two ALK overexpressions, two KRAS mutations and one BRAF mutation were detected. Apart from the EGFR mutation, the other driver gene mutations are less prevalent [2,3]; thus, the prevalence of actionable gene mutations is compatible with the literature.

In the era of digitalized PET systems, improved hardware and software now allow a larger image matrix size for PET image quantification [19]. However, most PET radiomic studies were derived from traditional analog PET units, which used smaller matrix sizes [5,22,24,25]. The radiomic features should be insensitive to matrix size changes, or the study results derived from the analog PET system cannot be generalized to the novel digitalized system. We found three popular ^18^F-FDG PET-derived features—SUV_max_ (a biomarker of glycolytic activity) and two volumetric features (MTV and TLG)—that are insensitive to matrix size changes. For glycolytic heterogeneity, we selected first-order entropy, which is one of the most robust features in respiratory motion, segmentation and reconstruction methods [16,17,18] for analysis. We found that the entropy was also robust for different matrix sizes. Our data provide evidence for the generalizability of entropy between traditional analog and novel digitalized PET systems. Additionally, because of the inherited lower image resolution of analog PET, the analyzable voxel number for small tumors (<10 mL) is limited. The lower number of analyzable voxels poses a concern for heterogeneity estimation in small tumors [26,27]. Nevertheless, we found that the first-order entropy of lower matrix-size images showed good consistency with that of higher matrix-size PET in small tumors (ICC = 0.894), indicating that the calculation of first-order entropy is not adversely affected by a low voxel number. Riegler et al. compared the quantitative ^18^F-FDG PET features between different matrix sizes. They reported a significantly higher SUV_max_ for images with higher matrices [20]. In our study, the SUV_max_ was also higher in the images of the standard matrix compared to the reduced matrix (12.1 ± 7.03 vs. 11.8 ± 7.00, *p* = 0.015). The lower SUV_max_ in the reduced matrix size image may result from partial volume averaging. Therefore, although the ^18^F-FDG PET-derived features may show excellent correlation between different image matrix sizes (high ICC), differences in the absolute value may still exist. We also analyzed the reliability of several other radiomic features and found that the higher-order features are more likely to be sensitive to image matrix size changes (Appendix A). Therefore, although changing the matrix size may not affect our study’s four reliable image features (SUV_max_, MTV, TLG and entropy), it may affect the measurement of some higher-order features.

The image-based glycolytic entropy exhibited a weak but significant positive correlation with Shannon entropy. As phenotypes are derived from genotypes, higher clonal heterogeneity may express a more heterogeneous phenotype. However, glycolysis is one of the many phenotypes expressed in the genome. Additionally, we could not determine the level of gene expression based on the gene mutation status. Additionally, the specimen for gene sequencing is only a part of the primary tumor, unlike the whole tumor featured in the image-based method. Therefore, although glycolytic entropy was positively and significantly correlated with Shannon entropy, the correlation was not strong. We also found that glycolytic entropy was significantly higher in patients with an advanced staging status, whereas genomic heterogeneity showed no significant difference. As tumor glycolysis is associated with oncogenic signaling pathway alterations [8,28], glycolytic features are associated with more aggressive behavior and metastatic potential [5,28]. Herein, the intensity, volumetric and heterogeneity glycolytic features were positively linked to higher cancer staging, suggesting that glycolysis is a biomarker of tumor aggressiveness. In contrast, Shannon entropy features the heterogeneity of the whole exome, which may include many other genes unrelated to oncogenesis. Therefore, a larger cohort may be required to demonstrate the association between Shannon entropy and cancer staging. Our preliminary survival analysis also showed that a higher TMB, SUV_max_ and entropy were associated with an unfavorable OS. Our results are compatible with the literature, where TMB and ^18^F-FDG PET-derived features were found to be predictive of survival outcomes in patients with lung cancer [5,29,30]. However, the follow-up period of our cohort was short and the patient characteristics were heterogeneous. Therefore, a study with a longer follow-up period and a uniform patient cohort is required to confirm our findings. Finally, Shannon entropy exhibited a significant negative correlation with TMB (r of approximately −0.55). Other studies have reported contradictory results regarding the association between tumor genomic heterogeneity and TMB, and no consensus has been reached [31,32]. The pathophysiology of tumor genomic heterogeneity and TMB is complex and the method for calculating the two biomarkers remains diverse [15,33]; therefore, the exact mechanism of the negative association between Shannon entropy and TMB is beyond the scope of this study and thus requires further research to unravel.

Although we found a weak and significant correlation between Shannon entropy and image-based entropy, the results were not corrected for multiple testing. When using the Bonferroni correction, the correlation was statistically insignificant (*p*-value should be <0.001 to be statistically significant after Bonferroni correction). As our preliminary results showed that the correlation between Shannon entropy and image-based entropy might be weak, our study cohort may be too small to show statistical significance after the Bonferroni correction. Therefore, this finding should be tested in a larger cohort.

We also found a significantly higher TMB in ever-smokers, which is consistent with the literature [34,35]. Another study suggested that cancers directly exposed to tobacco smoke (such as lung and laryngeal cancers) exhibit hypermutation, possibly attributed to DNA damage elicited by tobacco carcinogens [34]. In contrast, although TMB and PD-L1 immunohistochemistry are predictors of immunotherapy outcomes in NSCLC [36], these two biomarkers do not correlate with each other (r of approximately 0 in our study) [37]. PD-L1 is involved in the immune checkpoint pathway, the target of many immunotherapies [38], whereas the predictive value of TMB has been proposed to be established based on its relationship to neoantigens [15]. Therefore, the different mechanisms of PD-L1 expression and TMB may explain the poor correlation between these two biomarkers. Squamous cell lung cancer reportedly shows higher SUV_max_ on ^18^F-FDG PET compared with adenocarcinoma [25]. In our study, the mean SUV_max_ of squamous cell carcinoma was higher than that of adenocarcinoma, though no statistical significance was found (Appendix A; 14.1 ± 9.99 vs. 11.3 ± 5.81; *p* = 0.397). Our study population was not large; therefore, the difference in radiomic features between different histological types should be clarified further in a larger cohort.

Finally, low tumor purity results in a lower MAF of WES sequencing data, and the gene mutations for less populated clones may not be detected. Therefore, low tumor purity may affect the measurement of TMB and genomic heterogeneity features. Anagnostou KJ et al. showed that when the purity is less than 50%, TMB correlates with tumor purity, and they recommend correction of the TMB in cases where the tumor purity is less than 50% [39]. Other studies have suggested gene sequencing is able to detect mutations in samples with a purity of at least 20% [40,41]. In our study, we only included samples with at least 20% tumor purity, with a median of 50% (20–100%). Furthermore, in specimens with purities of <50%, the tumor purity did not correlate with TMB (r = −0.272 and −0.287 for TMBc and TMBm, respectively; *p* = 0.275 and 0.248, respectively) or genomic heterogeneity features (r = 0.207, 0.101, 0.277 and 0.156 for MATHc, MATHm, Shannon entropy.c and Shannon entropy.m, respectively; *p* = 0.409, 0.690, 0.266, and 0.537, respectively). The poor correlation between genomic features and tumor purity may suggest that lower purity has little effect on our genomic feature calculation. Additionally, not all studies on genomic features (including TMB, MATH and Shannon entropy) were corrected for tumor purity [13,42]. Therefore, the effect of tumor purity on the measurement of TMB and genomic heterogeneity needs further study to elucidate.

Our study had several limitations. First, the patient cohort was small. Second, although most clinical sequencing services adopt tumor-only algorithms for somatic mutation calls, the traditional filtering method is performed on tumors and matched normal DNA [15]. Thus, the reliability of the genomic features is ideal to be tested with matched normal samples. Third, the study population was heterogeneous and the follow-up period was short. Therefore, a more uniform cohort with a longer follow-up period is needed to confirm the association between radiogenomics and survival outcomes. Further studies with more uniform patient characteristics and a long-term follow-up to test the prognostic value of radiogenomic features would confirm and expand the clinical utility of radiogenomics in NSCLC.

## 4. Materials and Methods

### 4.1. Patients

Our prospective study was approved by the local institutions’ review council and the Ethics Committee (IRB108-249-A) and carried out under the Helsinki Declaration. This study was registered on ClinicalTrials.gov (identifier: NCT04314349). Our study data were collected between August 2020 and October 2022, and all the patients provided written informed consent. We included patients with a pathological diagnosis of NSCLC without prior cancer treatment. All participants underwent pre-treatment examinations, including contrast-enhanced computed tomography (CT) of the chest to the upper abdomen, ^18^F-FDG PET/CT and gadolinium-enhanced magnetic resonance imaging (MRI) of the brain. Our study included 46 patients and we have summarized their characteristics in Table 2. Pre-treatment evaluation and final cancer stage were discussed and determined at the Multidisciplinary Lung Cancer Conference directed by our thoracic oncology research team. We determined the stage according to the 8th edition of the American Joint Committee on Cancer (AJCC) manual [43]. The expressions of ALK and ROS-1 were assessed by immunohistochemistry (IHC) using Ventana D5F3 CDx and SP384 assays (Roche Diagnostics, Mannheim, Germany), respectively. The IHC of ALK is highly accurate in predicting the rearrangement of ALK on fluorescent in situ hybridization (pooled sensitivity and specificity of 0.97 and 0.99, respectively) [44]; therefore, we used the ALK IHC results to represent the status of ALK rearrangement. Positive ROS-1 IHC results were further tested with fluorescent in situ hybridization using a Vysis ROS1 Break Apart FISH Probe Kit (Abbott, Abbott Park, IL, USA) to confirm the presence of rearrangement. We also assessed the IHC of programmed cell death ligand 1 (PD-L1) using a Ventana SP263 assay (Roche Diagnostics, Mannheim, Germany) for every patient. PD-L1 expression TC was recorded in every patient [37,45].

### 4.2. Technique and Analysis of Whole Exome Sequencing

We selected primary tumor specimens with an at least 20% tumor cell content for WES [41]. We extracted tumor DNA from paraffin-embedded samples using a QIAamp DNA FFPE Tissue Kit (QIAGEN GmbH, Hilden, Germany). Tumor DNA started by shearing at least 300 ng of DNA through enzyme digestion (Illumina^®^ DNA Prep with Enrichment, San Diego, CA, USA) to obtain 150 base pairs fragments, which were then ligated to an adaptor for amplification. One hundred nanograms of the prepared library was hybridized with an illumine exonic probe (45.0 Mb, Illumina^®^ DNA Prep with Enrichment, Illumina, San Diego, CA, USA) for 2 h. After library preparation, all samples were sequenced using the NovaSeq platform (Illumina, San Diego, CA, USA) with 150 bp paired-end reads. The average coverage depth was 119×. The quality of read files (fastq) was assessed using FastQC (v0.11.8, Brabraham Institute, Brabraham, Cambridge, UK) and then aligned to the human reference UCSC hg19 (NCBI build 37.2) using the BWA software (v0.7.17, Wellcome Trust Sanger Institute, Wellcome Genome Campus, Cambridge, UK) to generate SAM format alignment files [46]. The files were converted and sorted into BAM format using SAMtools software version 0.1.19, and duplicate reads were marked using Picard Tools MarkDuplicates software (v2.20.2, Broad Institute, Cambridge, MA, USA). The processed BAM files were used as inputs for GATK (version 4.2.6.1, Broad Institute) to identify germline and somatic mutations. Annotated and filtered variants were manually checked using the Integrative Genomics Viewer (IGV, v2.8.9, Broad Institute) and then confirmed by Sanger sequencing [47]. Mutant variants with a depth of less than 20×, an MAF of less than 5% or those other than protein-coding missense mutations were filtered out [14,15,48]. Somatic mutation calls were determined using the PAF cutoff based on the GnomAD database (v2.1.1) for the general and Eastern Asian populations [49]. We tested PAF cutoffs of 1% and 0.01% for somatic mutation calls [14]. TMB and genomic heterogeneities from different PAF cutoff values were calculated.

TMB was defined as the number of somatic mutations per Mb, and TMBc and TMBm were calculated based on PAF cutoff values of 1% and 0.01%, respectively. The two genomic heterogeneity features in our study were MATH and Shannon entropy [12,13]. MATH was calculated as follows:MATH=100×MAD/median
where the median is the center of the distribution of the MAF and the MAD is the median of the absolute difference of each MAF from the median MAF [12]. MATHc and MATHm were based on PAF cutoffs of 1% and 0.01%, respectively.

Shannon entropy was calculated according to the distribution of probabilities (P(*f*)) of each MAF with a bin size of 10 [13], as follows:$$\mathrm{Shannon}\;\mathrm{entropy}=-\overset{10}{\underset{f=1}{\sum }}P\left(f\right)\log_{2}\left(P(f\right))$$

The Shannon entropy.c and Shannon entropy.m were based on PAF cutoffs of 1% and 0.01%, respectively.

### 4.3. ^18^F-FDG PET Imaging and Feature Extraction

Patients fasted for at least 6 h before intravenous injection of ^18^F-FDG (400 ± 10% MBq), and the blood glucose level should not have exceeded 200 mg/dL. We acquired the images 60 min after radiotracer injection using a 4-ring GE Discovery MI PET/CT system (GE Healthcare, Milwaukee, WI, USA). First, a transmission CT was acquired in the helical mode with a pitch of 0.984, and the slice thickness was 2.79 mm. The tube voltage was 120 kV and an automated tube current (range: 15–180 mA) was used. Then, we acquired the PET images from the vertex to the mid-thigh using the list three-dimensional mode. The scanning time was 150 s per table position (20 cm per table position with a 3 cm overlap for every contiguous frame). The PET was reconstructed using the Q.clear algorithm (β = 550) with CT-measured attenuation correction. Standard (256 × 256) and reduced (128 × 128) matrix sizes were used to generate standard (pixel size of 2.7 mm × 2.7 mm) and lower resolution (pixel size of 5.4 mm × 5.4 mm) image sets. The PET slice thickness was 2.79 mm.

We used PBAS 4.0 (PMOD Technologies Ltd., Zurich, Switzerland) for the segmentation of PET images. The SUVs of the PET images were normalized to the patient’s body weight and the administered radioactivity. An experienced nuclear medicine physician placed a volume of interest large enough to include the primary tumor. We then applied a 41% SUV_max_ threshold for segmentation [50]. We defined the segmented volume as the primary tumor MTV. The primary tumor TLG was the product of the MTV and SUV_mean_ (TLG = MTV × SUV_mean_). We selected first-order entropy, which is one of the most robust image features [16,17,18], to represent tumor glycolytic heterogeneity. The segmented volume of the primary tumor was used to compute the primary tumor entropy. We performed the computation on Pyradiomics 2.2.0 (Harvard Medical School, Boston, MA, USA) using a fixed bin size of 0.25 SUV [51,52]. Features extracted from Pyradiomics complied with the definition of the Image Biomarker Standardization Initiative [52], and the mathematical formula of PET-based entropy is as follows:$$\mathrm{Entropy}=-\overset{\mathrm{Ng}}{\underset{i=1}{\sum }}P\left(i\right)\log_{2}\left(P\left(i\right)+\varepsilon \right)$$
where P(*i*) is the probability of distinct resampled values and Ng is the total number of discrete intensity levels within the segmented volume. ε is an arbitrarily small positive number (≈2.2 × 10^−16^).

We summarized the procedure of radiogenomic feature extraction in Figure 4.

### 4.4. Statistical Analysis

Baseline patient characteristics were presented as proportions, frequencies, means or medians, as appropriate. We analyzed the reliability of WES-based genomic and ^18^F-FDG PET-derived radiomic features using ICC. Our ICC used a two-way random model, a single measurement type, to report the consistency of each radiogenomic feature. An ICC value less than 0.5 suggests poor reliability, a value between 0.5 and 0.75 means moderate reliability, a value between 0.75 and 0.9 indicates good reliability and a value greater than 0.9 indicates excellent reliability [53].

We also assessed the correlation between WES-based genomic features, ^18^F-FDG PET-derived radiomic features and clinical characteristics using Pearson’s correlation coefficient (r), *t*-tests or Mann–Whitney U tests, as appropriate. Patients were further followed up until death or November 2022, whichever occurred first. The OS was calculated from the date of diagnosis to the date of death or censored at the last follow-up for surviving patients. The association of the study variables with survival data was examined using Cox regression analysis. Statistical analyses were performed using SPSS software (version 20.0; SPSS Inc., Chicago, IL, USA). Statistical significance was set at a two-tailed *p*-value of <0.05.

## 5. Conclusions

Shannon entropy is a more reliable genomic heterogeneity feature than MATH. The ^18^F-FDG PET-derived first-order entropy was not sensitive to the image matrix size change. This may reliably bridge the study results from the old analog PET system to the novel digitalized PET units. Therefore, the entropy-based radiogenomic features are ideal biomarkers for research and further clinical use for NSCLC.

## Figures and Tables

**Figure 1 ijms-24-03988-f001:**
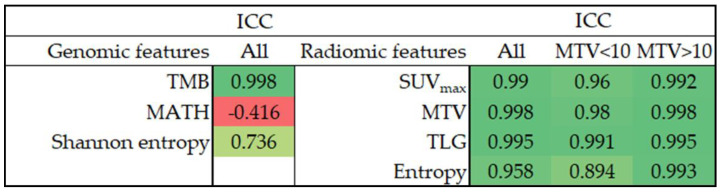
The ICC of radiogenomic features. ICC, intraclass correlation coefficient; TMB, tumor mutation burden; MATH, mutant-allele tumor heterogeneity; SUV, standardized uptake value; MTV, metabolic tumor volume; TLG, total lesion glycolysis.

**Figure 2 ijms-24-03988-f002:**
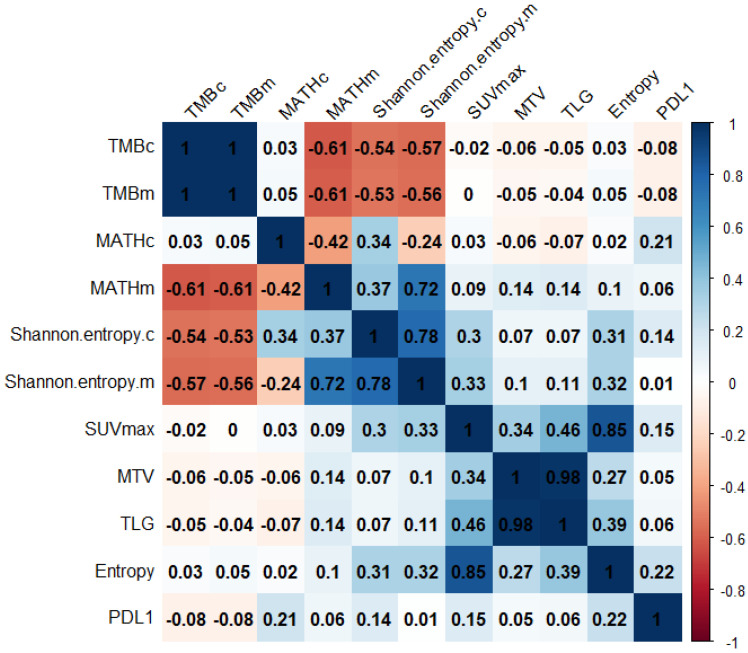
The correlation matrix of radiogenomic features. Values are Pearson’s correlation coefficients. TMB, tumor mutation burden; MATH, mutant-allele tumor heterogeneity; SUV, standardized uptake value; MTV, metabolic tumor volume; TLG, total lesion glycolysis; PDL1, programmed cell death ligand 1 expression level.

**Figure 3 ijms-24-03988-f003:**
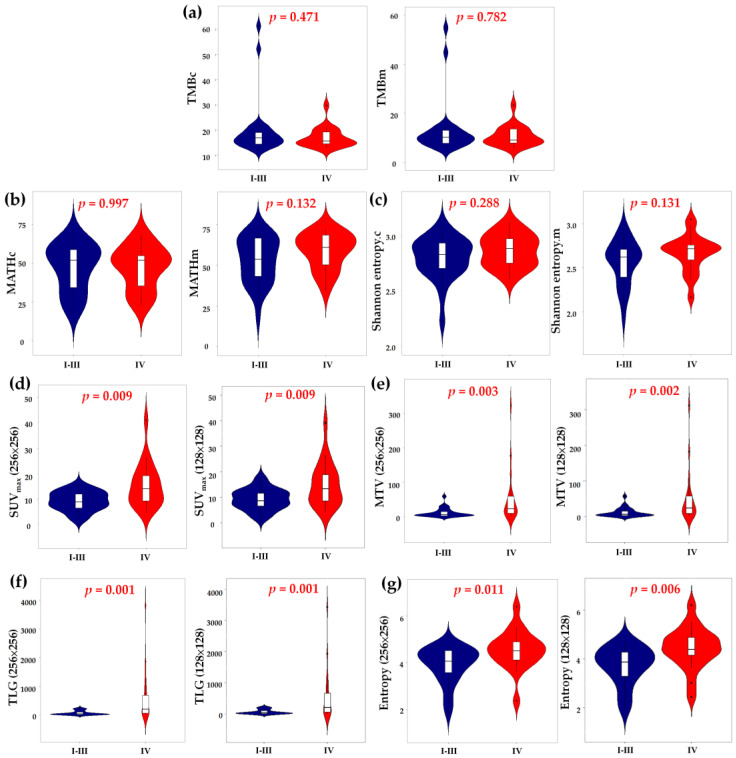
Comparing the TMB (**a**), MATH (**b**), Shannon entropy (**c**), SUV_max_ (**d**), MTV (**e**), TLG (**f**) and entropy (**g**) between patients with cancer stages of I to III and IV. TMB, tumor mutation burden; MATH, mutant-allele tumor heterogeneity; SUV, standardized uptake value; MTV, metabolic tumor volume; TLG, total lesion glycolysis.

**Figure 4 ijms-24-03988-f004:**
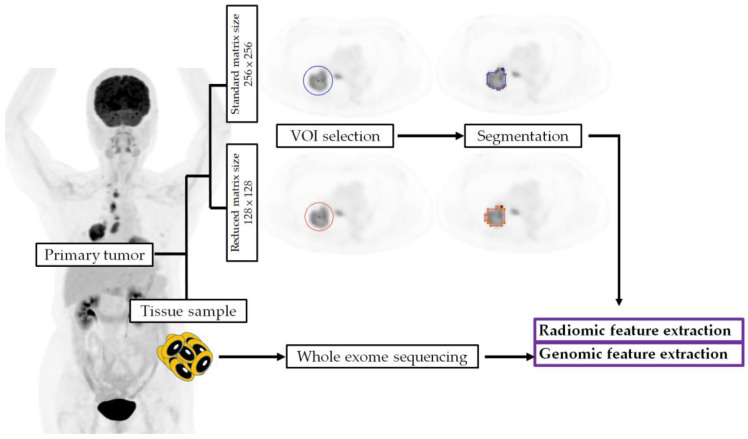
The procedures of radiogenomic feature extraction. VOI, volume of interest.

**Table 1 ijms-24-03988-t001:** The association of radiogenomic features with overall survival.

Variables	HR (95% CI)	*p*-Value
TMBc	1.048 (1.002–1.096)	0.042
TMBm	1.049 (1.003–1.096)	0.037
MATHc	1.045 (0.988–1.106)	0.121
MATHm	0.972 (0.929–1.016)	0.210
Shannon entropy.c	6.581 (0.082–530.160)	0.400
Shannon entropy.m	1.727 (0.116–25.830)	0.692
SUV_max_ (256 × 256)	1.502 (1.172–1.923)	0.001
SUV_max_ (128 × 128)	1.178 (1.066–1.301)	0.001
MTV (256 × 256)	1.003 (0.996–1.010)	0.400
MTV (128 × 128)	1.003 (0.996–1.010)	0.387
TLG (256 × 256)	1.000 (1.000–1.001)	0.339
TLG (128 × 128)	1.000 (1.000–1.001)	0.267
Entropy (256 × 256)	2.981 (0.927–9.590)	0.067
Entropy (128 × 128)	3.076 (1.019–9.286)	0.046

HR, hazard ratio; CI, confidence interval; TMB, tumor mutation burden; MATH, mutant-allele tumor heterogeneity; SUV, standardized uptake value; MTV, metabolic tumor volume; TLG, total lesion glycolysis.

**Table 2 ijms-24-03988-t002:** Patient characteristics in this study (*n* = 46).

Characteristics	Value
Age, years, mean (range)	67 (45–89)
Sex, *n* (%)	
Male	31 (67.4)
Female	15 (32.6)
Cigarette smoking status, *n* (%)	
Ever-smoker	27 (58.7)
Never used	19 (41.3)
Histology, *n* (%)	
Adenocarcinoma	35 (76.1)
Squamous cell carcinoma	11 (23.9)
T classification, *n* (%) ^a^	
1b	4 (8.7)
1c	10 (21.7)
2a	8 (17.4)
2b	5 (10.9)
3	7 (15.2)
4	12 (26.1)
N classification, *n* (%) ^a^	
0	23 (50.0)
1	1 (2.2)
2	12 (26.1)
3	10 (21.7)
M classification, *n* (%) ^a^	
0	26 (56.5)
1a	6 (13.0)
1b	3 (6.5)
1c	11 (23.9)
Overall stage, *n* (%) ^a^	
IA2	3 (6.5)
IA3	8 (17.4)
IB	5 (10.9)
IIA	1 (2.2)
IIB	4 (8.7)
IIIA	4 (8.7)
IIIB	1 (2.2)
IVA	9 (19.6)
IVB	11 (23.9)
EGFR mutation status, *n* (%)	
Wild type	30 (65.2)
Exon 19 deletion	7 (15.2)
L858R missense mutation	9 (19.6)
ALK overexpression, *n* (%)	2 (4.3)
ROS-1 rearrangement, *n* (%)	0 (0)
MET mutation, *n* (%)	0 (0)
BRAF V600E missense mutation, *n* (%)	1 (2.2)
KRAS G12C missense mutation, *n* (%)	2 (4.3)
PD-L1 expression, median (range)	1% (0–100%)
Initial treatments, *n* (%)	
Surgery	26 (56.5)
Chemoradiotherapy	2 (4.3)
Chemotherapy	4 (8.7)
Tyrosine kinase inhibitor	8 (17.4)
Immune checkpoint inhibitor	6 (13.1)

EGFR, epidermal growth factor receptor; PD-L1, programmed cell death ligand 1. ^a^, Stage according to the 8th edition of the American Joint Committee on Cancer manual.

## Data Availability

The data presented in this study are available on request from the corresponding author. The data are not publicly available due to privacy and ethical restrictions.

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
