# Peer review of "Genomic and Glycolytic Entropy Are Reliable Radiogenomic Heterogeneity Biomarkers for Non-Small Cell Lung Cancer"

_ijms, 2023, doi:10.3390/ijms24043988_

Round 1

Reviewer 1 Report

In their manuscript, the authors investigate the correlation of a number of radiogenomic features in non-small cell lung cancer. They especially focus on PET signal with different image matrix sizes and measures of genomic heterogeneity obtained in the absence of a non-tumor reference sample. Overall, the manuscript does not provide a wealth of new insights, but is useful in confirming that changes in matrix size do not seem to affect heterogeneity of PET imaging. Thus, its findings are relevant to clinical practitioners and researchers in the field of medical imaging. I do not, however, find the genomic analysis to be convincing.

Detailed comments:
- I assume that this may be the case due to the lack of follow-up (yet?) in the patient cohort, but merely using correlation measures leaves the main practical question unaddressed: are the values good predictors for patient outcome? Figure 3 does show the difference in tumor stage for both genomic cutoffs, but no such figure is provided for the image matrix sizes. The correlation is high enough that this seems like an obvious assumption, but then the same is true for the correlation and ICC of e.g. SUV, TLV, and MTV. If no outcome data are available, at least some numbers on staging would be appreciated.

- in figure 1, most of the imaging-based features are somewhat superfluous. If changing the image matrix size were to affect volumetric measurements beyond some uncertainty arising from discretized voxel space, this would call all PET imaging into question.

- I see no mention of correction for multiple testing.

- Looking at the formula, it is immediately obvious that MATH and Shannon entropy are dependent on tumor purity. Since this in turn may very well be correlated with tumor size and stage, and it does not seem to be controlled for, it is unsurprising that both values show high variability. It is not clear to me what value they add to the manuscript.

- It is good to see that the TMB is highly correlated independent of the cutoff chosen, but this does not show that a non-tumor reference sample is dispensable. For all we know, they may be highly correlated with each other, but correlate extremely poorly with the actual TMB in the patient.

- Some literature on matrix size in PET imaging is missing. They do metion that there are few studies, but don't cite the existing ones. For example, Riegler et al. report that increased matrix size can improve lung cancer detection, but may also increase the SUV (https://pubmed.ncbi.nlm.nih.gov/28583635/). This is not something that could be demonstrated by focusing on correlations alone, so I'm wondering whether the authors see similar effects in their raw data?

Minor comments:

- Some of the labels in figure 3 are hard to read.

Overall, the manuscript has some merit, albeit mostly in confirming intuitions about readiogenomic features. However, the genomic part is rather shaky, and the PET-based part adds releatively little information to what is known, especially considering that there are previous studies that investigated matrix size effects in PET imaging in lung cancer.

Reviewer 2 Report

      This  is a really interesting article, and the research results has high clinical significance. The research design is appropriate and methods are adequately described. I have the following question.

       As different histology type of NSCLC has a huge difference of 18F-FDG uptake,  can you explain if it affects your research, or can do further research?

      The line 85, for "entropy.m", there is a spelling error. Please check if there is any other minor spelling error.

Reviewer 3 Report

In this original study entitled "Genomic and glycolytic entropy are reliable radiogenomic heterogeneity biomarkers for non-small cell lung cancer" Yu-Hung Chen et al. present the comparison of the different genomic heterogeneity features - using the intra-class correlation coefficient (ICC) of tumor mutation burden, mutant-allele tumor heterogeneity, standardized uptake value, metabolic tumor volume total lesion glycolysis.

Although this is an exciting issue in oncology, the manuscript has several limitations:

1) the introduction has many shortcomings, and the part related to the gene test (or genetic testing) should be elaborately described -

f.ex. lines 40-43

Besides histopathology, novel gene tests enable more detailed cancer sub-classification and precise decisions [2-3]. Therefore, NSCLC is a heterogeneous disease, and gene testing has become pivotal in its management.  

Lines 45 -46 "The metabolic radiomics derived from 18F-FDG PET features tumor glycolytic activity and heterogeneity, which are associated with NSCLC prognosis" - How are those features associated with NSCLC prognosis?

2) Please rewrite the following sentence (lines 49-50)

"Tumor heterogeneity is linked to tumor evolution, aggression, and metastasis, and is thus associated with prognosis [9-11]" - How is it linked? Please give more details.

3) what are the glycolytic aspects? (lines 50-51) " Radiogenomics enables the quantification of tumor heterogeneity from both genomic and glycolytic aspects."

4) the study aimed to compare the "reliability of two commonly used genomic heterogeneity features (MATH and Shannon entropy) using different PAF cutoffs". The genomic heterogeneity features were taken from Whole Exome sequencing (WES). The WES results are described a little. The genomic features (regarding the highly mutated genes) could be described in more detail in the Result section and the Discussion.  

5) Table 1. Patient Characteristics in this Study - should be moved to the Materials and methods section. The age range should be added.

6) If there was a WES analysis, more data on the mutational status (of the most prominent genes in NSCLC) should be given. Also, the analysis - the comparison of the genomic features (presented in Table S1) could be performed depending on the mutation of the other genes.

7) Figure 1 - the horizontal bar stands for the correlation. 

8) In the results section (2.3), the PD-L1 expression pop-up, could You give more details on the PD-L1 expression range (maybe in Table 1).

9) Figure S2 - it is hard to evaluate which data are for the standard image matrix size and which are for the reduced. 

Round 2

Reviewer 1 Report

While I'm not fully convinced about the contribution of the manuscript to the field, the authors have addressed my criticisms from the previous round, albeit mostly through addition of limitation statements. Beyond this, I do not see major errors in need of correction.

Reviewer 3 Report

Thank You for introducing the proposed changes in the manuscript. In the current version, Yu-Hung Chen et al.'s manuscript "Genomic and glycolytic entropy are reliable radiogenomic heterogeneity biomarkers for non-small cell lung cancer" will be valuable for the broader scientific community.